

# Debris flow run-out simulation and analysis using a dynamic model

Raquel Melo[1], Theo van Asch[2], José L. Zêzere[1]

[1]Centre for Geographical Studies, Institute of Geography and Spatial Planning, Universidade de Lisboa, Edifício IGOT, Rua Branca Edmée Marques, 1600-276 Lisboa, Portugal

5  [2]Faculty of Geosciences, Utrecht University, P.O. Box 80115, 3508 TC Utrecht, The Netherlands

*Correspondence to*: Raquel Melo (raquel.melo@campus.ul.pt)

**Abstract.** Only two months after a huge wildfire occurred in the upper part of a valley located in Central Portugal, several debris flows were triggered by intense rainfall. The event caused infrastructural and economical damage, although no life was lost. The present research aims to simulate the run-out of two debris flows occurred during the event as well as to calculate by back-analysis the rheological parameters and the excess rain involved. Thus, a dynamic model was used, which integrates surface runoff, concentrated erosion along the channels, propagation and deposition of flow material. The rheological and entrainment parameters obtained for the most accurate simulation were then used to perform three scenarios of debris flows run-out at the basin scale. Due to the lack of quantitative information to validate these models, the results were compared with historical references of debris flow events in the study area. Six streams were identified, where debris flows occurred in the past and caused material damage and loss of lives. The worst-case scenario carried out at the basin scale shows a potential inundation that may affect 345 buildings at the present day.

**Keywords:** debris flows, dynamic modeling, run-out simulation, basin scale analysis.

## 1 Introduction

20  The debris flow initiation process can be subdivided into two main types of mechanism (e.g., Coussot and Meunier, 1996; Hungr, 2005; Van Asch et al., 2014): (i) infiltration-triggered soil slips, which develop into debris flows, and (ii) surface runoff erosion with entrainment of loose material. The latter is quite common in areas with scarce vegetation or that have been recently affected by forest fires, thus becoming more susceptible to erosion (e.g., Cannon et al., 2001a; Cannon and Gartner, 2005; Cannon et al., 2008; Santi et al., 2008; Baum and Godt, 2010; Cannon et al., 2011; Kean et al., 2013; Staley 25  et al., 2013b; Staley et al., 2014).

The consumption of vegetation, ash deposition, changes in physical properties of soils and rocks and the presence of water-repellent soils are typical consequences of fire (Cannon and Gartner, 2005; Cannon et al., 2010; Parise and Cannon, 2012). In addition, the hydrological response of a burned basin includes the decrease in the infiltration rate and consequently the increase of surface runoff (Cannon, 2001; Cannon and Gartner, 2005; Cannon et al., 2010). Therefore, burned basins are 30  strongly susceptible to debris flows, which may be very dangerous, especially for two reasons (Cannon et al., 2011): (i) the





amount of rainfall responsible for the triggering of debris flows may be lower when compared to areas that have not been recently affected by wildfires; and (ii) debris flows can be generated in places without past or historical occurrences.

The development of simulation techniques, especially in recent years, has allowed dynamic modeling to become an increasingly important tool for simulating the characteristics and behavior of debris flows. Nevertheless, the knowledge

about the initiation mechanisms, erosion and propagation of debris flows is still limited. While most models are focused on the propagation and deposition processes, only a few address the entrainment process and even less the initiation mechanism (Van Asch et al., 2014).

The dynamic approach is numerically solved through physically-based models based on fluid mechanics. Dynamic continuum models rely on the application of conservation laws of mass, momentum and energy, and the behavior of the flow

material is defined by its rheological properties (Dai et al., 2002; Quan Luna et al., 2012). One of the major challenges of the dynamic approach is to select the most appropriate rheology to simulate the flow behavior as well as to correctly estimate or calibrate the model's key parameters (e.g., O'Brien et al., 1993; Beguería et al., 2009; Hsu et al., 2010; Scheidl et al., 2013). Most frequently, these parameters are estimated by back-analysis of past events (e.g. Naef et al., 2006; Rickenmann et al., 2006; Hürlimann et al., 2008). Some of the rheological models frequently used to describe the behavior of debris flows are

(e.g., Naef et al., 2006; Beguería et al., 2009; Hungr and McDougall, 2009): 1) the Coulomb frictional resistance; 2) the Bingham model; 3) the Coulomb-Viscous; and (4) the Voellmy frictional-turbulent resistance.

Considering that dynamic models rely on physical-based assumptions, in most cases the validation is a complex task. Consequently, although widely applied at the slope scale, the dynamic approach has had little application at the medium scale. Regarding the latter, it stands out the work developed by Revellino et al. (2004) and Hürlimann et al. (2006) that

applied one-dimensional numerical models. More recently, Quan Luna et al. (2016) have implemented the "AschFlow", a two-dimensional one-phase continuum model that simulates the debris flow erosion and deposition processes. In this model, the debris flows are initiated by soil slips and the flow behavior is conditioned by the rheology.

In this context, the scarcity of studies applying dynamic models at the basin scale is evident. Thus, the main objectives of the present research are: 1) Modeling the run-out of two debris flows triggered during a rainstorm, only two months after a

25 wildfire affected the area; 2) To calculate by back-analysis the rheological and entrainment parameters, as well as the excess rain involved in the triggering of these two debris flows; 3) To apply a two-dimensional one-phase continuum model to estimate the parameters referred in (2). This model simulates the debris flow initiation process by surface runoff, concentrated erosion along the channels, propagation and deposition of flow material; 4) To perform three scenarios of debris flows run-out at the basin scale, using different values of excess rain for each scenario and considering the rheological

and entrainment parameters obtained for the most accurate simulation.





## 2 Study area

The study area is the upper part of the Zêzere valley located in the highest Portuguese mountain, the Estrela Mountain, in Central Portugal (Fig. 1). The elevation ranges from 668 m to 1990 m a.s.l. and gradually decreases from SSW to NNE. The geology is dominantly granitic (Migoń and Vieira, 2014): porphyritic medium- to coarse-grained granite and non-porphyritic

5 medium- to fine-grained granite outcrop over 73% of the study area. Hornfels constituting a metamorphic aureole are present in the northern part of the study area, whereas the bottom of the valley is covered by Quaternary deposits of fluvial and glacial origin.

The climate is Mediterranean, with warm and dry summers. The mean annual precipitation is circa 2500 mm in the Estrela Mountain summit (Vieira et al., 2004) and the wet season typically lasts from October to May.

## 3 Data and methods

### 3.1 The event of 30 October 2005 and historical events

During the summer of 2005 the southern part of the study area was affected by a large wildfire (Fig. 2) and the unprotected slopes were strongly affected by rainfall-triggered debris flows during an event registered on 30 October 2005. A total of 34

debris flows were identified most of them along the Eastern slope of the valley. Although no victims were registered, the national highway linking the main village to the most touristic places in the higher mountain was closed due to the debris flow occurrence (Melo and Zêzere, 2017). The debris flows were identified and mapped by aerial photo interpretation, analysis of morphological features from post-event topography (1:10,000 scale) and systematic field surveying with DGPS during 2011.

The first records on debris flows, occurred in the study area, date back to the 19[th] century when a particular event registered in 1804 (occurred along the stream marked in green, in Fig. 2) generated 20 dead people (Melo and Zêzere, 2017). More recently, in 1993 a debris flow occurred in an area affected by a wildfire two years before (along the stream marked in pink in Fig. 2) and produced high material losses in a hotel. The remaining streams, shown in Fig. 2, were also affected by debris flow activity in the past. We think these events are underestimated, especially along the Zêzere valley, due to underreport

related to the scarce number of elements at risk. Indeed, there is a lack of quantitative information about debris flows that occurred in the study area. For instance, even for the latest ones, there is no record on the volume of the deposited material or even the total amount of rainfall, although we know it happened after a severe rainstorm.





## 3.2 Model description

The two-dimensional one-phase continuum model used was developed and applied by Van Asch et al. (2014). The model integrates, in a simple way, four components: surface runoff, erosion of loose sediments deposited along the channels, flow propagation and deposition.

It must be kept in mind that physical-based models always need to be adapted to the reality under study. In the research developed by Van Asch et al. (2014), the lack of information about the type of soils in the catchments led to the choice of a simple runoff model with a constant infiltration capacity, where the effects of initial moisture content and the sorptivity of the soil are ignored (Eq. 1):

$$h_r = (i - k_s),  \qquad (1)$$

Where $h_r$ is the excess rain (m s$^{-1}$) that will feed the surface runoff; $i$ is the rainfall intensity (m s$^{-1}$); and $k_s$ is the soil infiltration capacity (m s$^{-1}$).

The model integrates the entrainment process through a simple erosion module, which intends to reproduce several
mechanisms, such as lateral and vertical erosion, bed destabilization by infiltrating water and undrained loading (Van Asch et al., 2014). The abovementioned processes are related to flow velocity and flow above a critical height, according to the equations proposed by Eglit and Demidov (2005) and McDougall and Hungr (2005) (Eq. 2):

$$h_{sc} = \beta v \, (h - h^*),  \qquad (2)$$

Where $h_{sc}$ is the erosion rate (m s$^{-1}$); $\beta$ is the erosion factor (m$^{-1}$); $v$ is the velocity (m s$^{-1}$); $h$ is the flow height (m); and $h^*$ is the critical height for erosion to occur, which is arbitrarily assigned the value of 0.1 m (Van Asch et al., 2014). Given the lack of detailed information about the characteristics of the channels, no distinction is made between the erosion along the channels and along the slopes.

At each time step, the solid material is added to the flow until the maximum volumetric concentration is equal to 0.6. When the volumetric concentration of the solids is below the arbitrary limit of 0.2, the flow velocity is calculated using the Manning´s equation (Van Asch et al., 2014) (Eq. 3):

$$v = \frac{h^{2/3} sin\theta^{1/2}}{n},  \qquad (3)$$

Where $\theta$ is the slope angle in the direction of the steepest slope; and $n$ is the Manning´s coefficient (m$^{-1/3}$ s). When the volumetric concentration is higher than 0.2, a simple equation of motion is applied (Van Asch et al., 2014) (Eq. 4):



$$\frac{\partial v}{\partial t} = g\left[sin\theta\, cos\theta - k\, tan\theta - S_f\right], \tag{4}$$

Where $g$ is the gravitational acceleration (m s$^{-2}$); $k$ is the lateral pressure coefficient; and $S_f$ is the resistance factor, which depends on the flow rheology.

The routing of solids and water are separately, obeying the law of mass conservation, and for each time step a new concentration is calculated (Van Asch et al., 2014). In order to maintain the numerical stability of the model during the simulation, a flexible time step based on the Courant-Friedrichs-Lewy (CFL) condition is used (Beguería et al., 2009). The model is implemented in the PCRaster environmental modeling language (Karssenberg et al., 2001).

**3.3 Model setup**

The calibration of the model was performed by trial and error for the two largest debris flows triggered during the event of 2005. These two debris flows were selected based on its size and volume of the mobilized material, as well as the conservation of deposits at the time of field surveying. Due to the lack of quantitative information for validation purposes (i.e., thickness and volume of the deposits), we considered that the outputs from the model could be valid if they positively

answered all the following criteria (Fig. 3): a) The modeling results must reveal an agreement between the maximum run-out distance simulated and the maximum run-out distance observed; b) The simulation must mimic the deposition of material, with a few centimeters in thickness, observed along the debris flow transport zone; c) The maximum absolute thickness of the deposits in the accumulation area must not exceed 3.5 m and the mean value must be between 1.5 m and 2.0 m, as we registered during field work.

**3.3.1 Rain duration and excess rain values**

The duration and amount of rainfall involved in the triggering of debris flows are of major importance for the run-out assessment using a dynamic model. It is also known that rainfall hourly data are more important than rainfall cumulative daily data. For instance, Malet and Remaître (2011, in Van Westen et al., 2014) concluded that debris flows are generally triggered by storms lasting between 1 and 9 hours. However, there is no information about the duration or the total amount

of rainfall occurred in the mountain, where the debris flows were triggered. Nevertheless, the two nearby rain gauge stations registered a 2-hour period with the highest values of rainfall (Table 1). Although the two stations are located at distinct topographic positions and several kilometers away from the place where the debris flows occurred (Station A is located at the bottom of the mountain and Station B is in Manteigas village), the models were computed considering a rain duration of 2 hours, since we do not have more precise information.

Due to the uncertainty regarding the amount of rainfall occurred, as well as to the lack of detailed information about the soils in the catchments, we chose to calibrate the model based on the excess rain values, which means the amount of water that



will feed the surface runoff once the infiltration is exceeded. Thus, surface runoff reflects not only the total rainfall, but also the interaction between rainfall and the soil system.

### 3.3.2 Rheology and erosion coefficient

The calibration of the geotechnical parameters was performed using four different rheological models: 1) the Coulomb frictional resistance; 2) the Bingham model; 3) the Coulomb-Viscous; and (4) the Voellmy frictional-turbulent resistance. The Coulomb frictional resistance is based on the relation between the effective bed and normal stress at the base of the flow and the pore fluid pressure (Hungr and McDougall, 2009; Ferrari et al., 2014):

$$S_f = \tan \varphi', \tag{5}$$

$$tan\, \varphi' = (1 - r_u)\, tan\, \varphi, \tag{}$$

Where $S_f$ is the unit base resistance; $r_u$ is the pore-pressure ratio (-); and $\varphi$ (°) is the dynamic basal friction angle. The basal stress is frictional if $r_u$ is considered constant, which means that the total normal stress and the shear stress remain proportional. Thus, the equation can be simplified by only including the basal friction angle.

The Bingham model assumes that the debris flow material exhibits a visco-plastic behavior, with laminar flow. The basal shear stress is calculated using Eq. 6 (Beguería et al., 2009):

$$S_f = \frac{1}{\rho g h}\left(\frac{3}{2}\,\tau c + \frac{3\eta}{h}\,v\right), \tag{6}$$

Where $S_f$ (-) is the unit base resistance; $\rho g h$ is the normal stress ($\rho$ is the mass density (kg m$^{-3}$) of the flow and $g$ (m s$^{-2}$) is the gravitational acceleration); $\tau c$ (kPa) is the constant yield strength due to cohesion; $\eta$ (kPa s) is the dynamic viscosity; $h$ (m) is the thickness of the flow; and $v$ (m s$^{-1}$) is the flow velocity.

Since debris flows are often composed of sediments with different sizes, one way of considering the friction effect resulting from the contact between particles is to complement the Bingham model with the Coulomb´s friction resistance component (De Blasio, 2011). Thus, Johnson (1970, in De Blasio, 2011, among others) modified the Bingham model by incorporating the frictional resistance component, which led to the development of Coulomb-Viscous model, whose application extends to a wider range of fluids. In this model (Eq. 7), the yield strength results from the combination of cohesion and friction forces (Beguería et al., 2009):

$$S_f = tan\, \varphi' + \frac{1}{\rho g h}\left(\frac{3}{2}\,\tau c + \frac{3\eta}{h}\,v\right), \tag{7}$$



The Voellmy model was initially applied to the simulation of snow avalanches (Voellmy, 1955 in Ferrari et al., 2014) but ever since has been used for modeling landslides of granular material, cohesionless, with or without interstitial fluid (Ferrari et al., 2014) and with turbulent behavior. The model combines a basal friction coefficient similar to Coulomb´s apparent friction ($\varphi'$) and a resistance term (turbulent coefficient, $\xi$) similar to the Chézy resistance for turbulent water flows in open

channels. The basal shear stress is given by Eq. 8 (Ferrari et al., 2014):

$$S_f = \left[ tan\,\varphi' + \frac{v^2}{\xi h} \right], \tag{8}$$

Where $\xi$ (m s$^{-2}$) is the turbulent coefficient.

The range of values selected to represent the cohesion, the viscosity and the turbulent coefficient were taken from a compilation of studies carried out by Quan Luna (2012) about debris flows occurred in a similar geological context, namely in decomposed granites.

The erosion coefficient was calibrated according to the modeling results.

### 3.3.3 Constant parameters

In addition, the model requires the input of information (related to maps and some model parameters), which was considered constant during the calibration of the excess rain, rheology and erosion coefficient, namely:

   a)  Digital elevation model (DEM);
   b)  Soil thickness;
   c)  Manning's $n$ (= 0.04);
20 d)  Lateral pressure coefficient (= 1);
   e)  Gravitational acceleration (= 9.8 m s$^{-2}$);
   f)  Unit weight of water (= 10 kN m$^{-3}$);
   g)  Unit weight of the debris flow (decomposed granite + water = 19 kN m$^{-3}$);
   h)  Unit weight of the bed material (solid granite = 26 kN m$^{-3}$).

### 3.4 Debris flow run-out modeling at the basin scale using the two-dimensional one-phase continuum model

One of the main objectives of this research is to use the dynamic model to simulate the debris flow run-out at the basin scale. To achieve this purpose, three different scenarios were created based on the combination of excess rain values, erosion coefficient and rheological parameters that best reproduced the two major debris flows occurred in 2005. However, it is

30 necessary to keep in mind that these parameters - not only the precipitation, and consequently the surface runoff, but also the





rheology - most certainly vary over the basin. Thus, the creation of scenarios aims to address the following question: "what would be the response of the basin if a certain value of excess rain occurred, considering the rheological parameters and the erosion coefficient that best reproduced the run-out of the two major debris flows triggered during the 2005 event?". To answer this question, we developed three scenarios (A, B, and C), with excess rain values of 30, 35 and 40 mm h$^{-1}$,

respectively.

## 4 Results and discussion

The two maps used as inputs for the model are: (1) a DEM with a resolution of 5 m, which reflects the topography prior to the occurrence of the debris flows (Fig. 4); and (2) a map of the soil thickness – interpreted as the depth to bedrock – based

on the simplified geomorphologically indexed soil thickness (sGIST) model (Catani et al. 2010; Segoni et al. 2012) (Fig. 5). The latter was validated by comparing the results with 38 field point measures, resulting in a mean absolute error of 29 cm. It is important to emphasize the numerical and spatial limitation of these data. However, it is very difficult to find cuts in natural slopes along the study area, since most of them had some kind of anthropogenic intervention (mainly for roads construction).

The calibration of the rheology and the erosion coefficient is only performed after a minimum value of excess rain with capacity to trigger debris flows has been determined for the study area. We found out that excess rain values lower than 28 mm h$^{-1}$ do not generate debris flows. Thus, the calibration was performed using an amplitude of 1 mm h$^{-1}$. Regarding the calibration results for the two debris flows (DF#1 and DF#2, see Fig. 3), we also found that values above 30 mm h$^{-1}$ are required for sediment mobilization. For instance, DF#1 requires an excess rain of 32 mm h$^{-1}$ to allow the calibration of the

rheology and the erosion factor, since lower values result in insufficient deposit thickness and insufficient run-out distances, despite the variation of the previously mentioned parameters. DF#2 requires an excess rain of 33 mm h$^{-1}$ to allow the calibration of the rheology and the erosion factor.

As already referred, the range of values selected to calibrate the cohesion, the viscosity and the turbulent coefficient were taken from a compilation of studies carried out by Quan Luna (2012). The cohesion was calibrated using values between 0.8

and 1.0 kPa (with a range of 0.1 kPa). For the calibration of viscosity, we also used values between 0.8 and 1.0 kPa s (with a range of 0.1 kPa s). In the Voellmy model, the values used for the turbulent coefficient were between 400 and 2000 m s$^{-2}$ (with a range of 200 m s$^{-2}$). Concerning the apparent friction angle we choose the values of 9°, 14° and 21° which correspond to the minimum, medium and maximum values used in the studies compiled by Quan Luna (2012). The selection of only three values of apparent friction angle was intentional, in order to limit the number of simulations computed. Regarding the

erosion coefficient, values between 0.0010 and 0.0014 m (with a range of 0.0001 m) were tested.

Considering the validation criteria previously defined, the model versions using the Coulomb frictional and the Voellmy frictional-turbulent resistance were excluded. Looking at some modeling results (Fig. 6), it is clear that both rheologies



originate a material deposition only in the most distal part of the debris flows, which means the models are unable to reproduce the deposition observed along the transport zone.

The simulations performed with the Bingham and Coulomb-Viscous models do not present the abovementioned limitation. Tables 2 and 3 show the statistical summary for the 180 simulations performed with Bingham and Coulomb-Viscous models for DF#1 and DF#2. In general, both models apparently produce very similar results. Regarding DF#1, the main differences are related with the run-out, whose values are higher with the Coulomb-Viscous model. In relation to DF#2, the differences produced by both models are slightly more pronounced, standing out the maximum volume and the minimum run-out obtained with the Bingham model. In order to understand if there is a relation between some of the most important parameters that reflect the magnitude and intensity of debris flows, correlations between the maximum volume and the run-out, as well as the maximum volume and the maximum flow velocity are analyzed, considering the results obtained with both Bingham and Coulomb-Viscous models. The correlations established are summarized in Fig. 7 and 8. DF#1 presents a strong positive linear correlation between the maximum volume and the run-out (a) and the maximum flow velocity (b) ($R^2 =$ 0.90 and $R^2 = 0.95$, respectively) (Fig. 7). However, considering the relation between the maximum volume and the run-out, we can detect a few outliers in the results with Bingham model (Fig. 7, a). These outliers represent simulations that produced similar volumes but shorter run-outs, in comparison with the simulations performed with the Coulomb-Viscous model, despite the same values of cohesion, viscosity and erosion coefficient were used. The $R^2$ estimated for the correlation between the maximum volume and the maximum flow velocity (Fig. 7, b) indicates a strong positive linear correlation ($R^2 =$ 0.95) without major differences between the two rheological models. Regarding DF#2, the logarithmic function is the one that best describes the relation between the maximum volume and the run-out (Fig. 8, a), with a high coefficient of determination ($R^2 = 0.82$). However, for the same run-out, the higher volumes are observed with Bingham rheology. The relation between the maximum volume and the maximum flow velocity (Fig.8, b) is best represented by a positive linear correlation ($R^2 = 0.84$). The maximum velocities obtained are slightly higher when the Bingham rheology is used.

Considering that the previously analyzed models represent the material deposition along the transport zone, it must be verified if the other two criteria are also valid, namely the thickness observed in the accumulation area and the agreement between the maximum run-out distance simulated and the maximum run-out distance observed. The overall selection accounted 1 simulation with the Bingham model and 11 simulations with the Coulomb-Viscous model that had positively answered these two validation criteria.

Table 4 shows the combination of rheological parameters and erosion coefficient, as well as the maximum flow velocity, deposit thickness, volume and run-out estimated for the 12 most accurate models (the models computed for DF#1 are highlighted in blue and for DF#2 in orange). The debris flow run-out was estimated based on the distance between the initiation and the most distal position of the deposited material. The real run-out of DF#1 and DF#2 is respectively 521.8 m and 498.9 m. The maximum and the mean thickness of the deposits, as well as the volume, are slightly higher for DF#2, thought the velocity is lower. Regarding the run-out distances, the mean values vary 12 meters from the real one for DF#1 and 20 meters for DF#2. However, we highlight that the maximum run-out distance was delimited considering the abrupt



terminal lobes, composed of coarse material, whose deposits tend to remain preserved. But frequently, the head of the debris flows is overcome by the thin, saturated debris that constitute the body and the tail of the flow. So, in this case, since the field surveying was carried out six years after the event took place, it is possible that the real run-out distance was underestimated.

Fig. 9 shows the temporal evolution of the deposited material thickness considering a simulation whose results agree with all the three predefined criteria (Model run #62 for DF#1 and Model run #64 for DF#2). Compared to static models, the dynamic models have the advantage of allowing to simulate the evolution in space and time of a given process. For example, according to the models performed, it is possible to determine that the stages of initiation, transport and deposition of DF #1 had a total duration of 53 minutes, whereas DF#2 occurred over a period lasting 77 minutes.

The debris flow run-out modeling at the basin scale included the computation of three different scenarios considering 30, 35 and 40 mm h$^{-1}$ excess rain, during a period of 2 hours. The values assigned to the rheological parameters, as well as to the erosion coefficient, were based on the arithmetic mean obtained for the 12 most accurate simulations (shown in Table 4): $\varphi' = 14°$; cohesion = 0.9 kPa; viscosity = 0.9 kPa s; and $\beta = 0.0013$ m.

Fig. 10, 11 and 12 show the modeling results for scenarios A, B and C with excess rain values of 30, 35 and 40 mm h$^{-1}$,

respectively. Table 5 summarizes the maximum velocity, maximum thickness, total volume and maximum run-out obtained for the three scenarios. In a first analysis, the values obtained for scenarios A and B appear to be realistic, whereas scenario C has a much higher maximum thickness compared to the previous ones.

In Scenario A (Fig. 10) the debris flows achieve a maximum velocity of 3.5 m s$^{-1}$ and a maximum thickness of 4.2 m. Table 6 shows that 1% of the deposits have thicknesses equal or above 2 m. The overlay between the present-day buildings and the

modeling result shows that there are no buildings at risk. Moreover, as it can be seen in the enlargement frames in Fig. 10, only 5 out of the 34 debris flows triggered during the 2005 event were reproduced, which means the excess rain value used in this scenario (30 mm h$^{-1}$) is not high enough to mobilize the sediments along the gullies where the remaining debris flows occurred.

Concerning the Scenario B (Fig. 11), where an excess rain of 35 mm h$^{-1}$ was used, the debris flows achieve a maximum

velocity of 6.8 m s and a maximum thickness of 6.6 m. In this scenario, 1.6% of the deposits have thicknesses equal or above 2 meters (Table 7). The overlay between the present-day buildings and the simulation result accounts for a total of 116 buildings at risk. With the Scenario B, all the 34 debris flows triggered during the 2005 event were reproduced in the simulation. This is not entirely legible in Fig. 11 due to the overlay between the debris flows and the simulation result, but the frame (with green outline) shows an example. The total amount of debris flows simulated in this scenario surpasses the

event of 2005, which means the latter must lie between scenarios A and B. In addition, by comparing the modeling result with Fig. 2, it is clear that Scenario B is able to simulate the material deposition in 4 streams with historical reports of debris flow activity in the study area. In Fig. 11, the two frames outlined in black and red show the overlay between the streams with historical debris flow activity and the modeling result. The red frame shows the stream where the deadliest debris flow, known to date in the study area, took place killing around 20 persons and destroying a same number of houses. Considering





the simulation result we can, nowadays, account for 35 buildings at risk only in this stream. The black frame shows the stream in which was triggered the debris flow that affected a hotel in 1993. In this more recent event, eyewitnesses recalled debris deposits of about 1 meter against a wall of the hotel on the ground floor. In Scenario B, this hotel is also affected by a debris flow, but the model simulates a maximum flow thickness of 0.6 meters. However, it must be kept in mind that the

5      model is not considering any kind of obstacles that are not covered by the DEM.

Finally, Scenario C (Fig. 12), where an excess rain of 40 mm h$^{-1}$ was used, was considered the worst-case scenario of this study. Here, 2% of the deposits have thicknesses equal or above 2 m (Table 8) and we can also account for 345 buildings at risk. Moreover, this scenario is able to simulate the material deposition in all the 6 streams identified with historical debris flow activity in the study area.

## 5 Concluding remarks

In this research, we applied a two-dimensional one phase continuum model to calibrate, by back-analysis, the excess rain involved as well as the rheological parameters that best reproduced the two major debris flows triggered during an intense rainstorm in 2005, only two months after the area was affected by a huge wildfire.

The model applied simulates the debris flow initiation process by surface runoff, the erosion, the flow propagation and the deposition of the traveled material, which means the model is suitable to simulate debris-flows in recently burned areas, considering the main initiation mechanism is due to runoff erosion. Moreover, the erosion module allows to simulate the volume increase due to material entrainment. The rheological parameters and the erosion coefficient that produced the best simulations were used to compute a debris flow run-out modeling at the basin scale. Three scenarios were elaborated using

different excess rain values. Since these models are extremely difficult to validate in the study area, a comparison has been made with some historical data concerning the occurrence of debris flows in the river basin in question. Finally, for each of the scenarios, the elements at risk were accounted.

The debris flows run-out modeling using a physically-based model presents several advantages when compared with the most common data-driven models. Unlike the latter, physically-based approach allows for the estimation of flow velocities,

thickness of the deposits and impact force against obstacles. Such parameters are of paramount importance for the development of warning systems and structural mitigation measures. In addition, this type of model is not dependent on local conditions and do not require landslide inventories for modeling procedures, which is an advantage when the modeling is performed for locations where no inventories are available. Nonetheless, such inventories may remain essential to calibrate the rheological parameters by back-analysis and for validation purposes (Oliveira et al., 2017).

On the other hand, the reliability of the modeling results is strictly dependent on the DEM and the soil thickness map. When the modeling intends to reproduce an event occurred in a certain catchment, the DEM must represent the topography previous to the event. Plus, the soil thickness must be estimated for the entire catchment, which is not an easy task. The difficulty in estimating the spatial variation of this parameter frequently leads to the use of a constant value. However, this is





not advisable when the model is computed at the catchment scale and if the primary debris flow initiation mechanism is due to runoff erosion, unlike the located infiltration-triggered soil slips that afterwards develop into debris flows. If more precise information exists for the study area, namely the characteristics of the soils in the catchments and the hourly total amount of rainfall, the model can use these parameters instead of being calibrated based on excess rain. This will allow estimating

critical rainfall thresholds for a given area (Van Asch et al., 2014). As already highlighted by Quan Luna et al. (2016), the integration of different rheological models allows the comparison of the different results. This provides a flexible choice for the user, not only regarding the scenario that best reproduces the event occurred, but also concerning the use of the rheology most adequate to the type of event under analysis (e.g., hyperconcentrated or granular), which is a major advantage for the development of future scenarios.

The work developed also intends to show the importance of dynamic modeling for the debris flow run-out assessment. The outputs of the model allowed the calculation of flow velocity, thickness of the deposits, volume and extend of traveled material, at the basin scale. These are extremely important parameters that should be considered in further studies on hazard and risk assessment.

Despite many uncertainties, another advantage of dynamic modelling is the possibility to bring in the temporal component

for a hazard analysis for debris flows at a regional scale. Running scenario's with different intensities of rain events, which have different return periods, enables one to distinguish areas at risk, with different temporal frequencies and impact intensities. This can provide important information for cost benefit analyses and mitigation planning.

**Author contribution**

R. Melo, Th. W. J. van Asch and J. L. Zêzere conceptualized the study. Th. W. J. van Asch developed the dynamic model

and contributed for its adaptation to the study area. R. Melo and J. L. Zêzere performed field work. R. Melo prepared the data, the models and the manuscript with contributions from all authors.

**Competing interests**

The authors declare that they have no conflict of interest.

**Acknowledgements**

This work was financed by national funds through FCT—Portuguese Foundation for Science and Technology, I.P., under the framework of the project FORLAND—Hydro-geomorphologic risk in Portugal: driving forces and application for land use planning (PTDC/ATPGEO/1660/2014). Topography data courtesy of the Manteigas Municipality.



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





**Table 1.** Rainfall registered on 30 October 2005 at the two rain gauge stations near the Zêzere valley. Stations A and B are located at a linear distance of, respectively, ca. 8 km and 8.5 km from the place where debris flows occurred

| Hour | Rainfall (mm) | |
| --- | --- | --- |
| | Station A (719 m a.s.l.) | Station B (800 m a.s.l.) |
| 0:00 | 2.2 | 2.0 |
| 1:00 | 1.0 | 4.0 |
| 2:00 | 3.3 | 4.0 |
| 3:00 | 2.8 | 1.0 |
| 4:00 | 1.8 | 0.0 |
| 5:00 | 0.7 | 1.0 |
| 6:00 | 0.4 | 0.0 |
| 7:00 | 3.4 | 1.0 |
| 8:00 | 8.4 | 3.0 |
| **9:00** | **11.0** | **11.0** |
| **10:00** | **16.0** | **12.0** |
| 11:00 | 2.1 | 0.0 |
| 12:00 | 10.5 | 0.0 |
| 13:00 | 1.3 | 3.0 |
| 14:00 | 0.2 | 5.0 |
| 15:00 | 0.3 | 10.0 |
| 16:00 | 0.1 | 1.0 |
| 17:00 | 7.1 | 0.0 |
| 18:00 | 1.2 | 0.0 |
| 19:00 | 2.7 | 1.0 |
| 20:00 | 0.2 | 0.0 |
| 21:00 | 0.0 | 0.0 |
| 22:00 | 0.1 | 0.0 |
| 23:00 | 0.0 | 0.0 |
| Total | 76.8 | 59.0 |





**Table 2.** Statistical summary for the 180 simulations performed with Bingham and Coulomb-Viscous models for DF#1

|  |  | Max. velocity (m s) | Max. thickness (m) | Volume (m$^3$) | Run-out (m) |
|---|---|---|---|---|---|
| Bingham model | Maximum | 2.6 | 4.7 | 2736.1 | 585.9 |
|  | Minimum | 0.2 | 0.8 | 338.3 | 502.0 |
|  | Mean | 1.4 | 2.8 | 1400.2 | 520.8 |
|  | STD | 0.8 | 1.3 | 898.4 | 20.3 |
| Coulomb-Viscous model | Maximum | 2.7 | 5.0 | 2730.1 | 607.5 |
|  | Minimum | 0.1 | 0.4 | 322.0 | 475.0 |
|  | Mean | 1.4 | 3.0 | 1457.1 | 554.3 |
|  | STD | 0.8 | 1.5 | 866.3 | 39.2 |



**Table 3.** Statistical summary for the 180 simulations performed with Bingham and Coulomb-Viscous models for DF#2

| | | Max. velocity (m s) | Max. thickness (m) | Volume (m³) | Run-out (m) |
|---|---|---|---|---|---|
| Bingham model | Maximum | 3.5 | 5.0 | 11413.0 | 546.5 |
| | Minimum | 0.5 | 1.9 | 760.7 | 507.5 |
| | Mean | 1.6 | 3.8 | 3304.9 | 532.3 |
| | STD | 0.6 | 0.8 | 2540.2 | 15.7 |
| Coulomb-Viscous model | Maximum | 2.9 | 4.8 | 9057.6 | 546.5 |
| | Minimum | 0.2 | 0.5 | 593.2 | 464.9 |
| | Mean | 1.0 | 3.1 | 2107.2 | 519.3 |
| | STD | 0.4 | 0.9 | 1497.8 | 15.6 |





**Table 4.** Parameters used and estimations obtained for the most accurate simulations of DF#1

| Simulation number | β (m) | $\varphi'$ () | Cohes. (kPa) | Visc. (kPa s) | Max. Vel. (m s) | Vol. ($m^3$) | Max. thickn. (m) | Mean thickn. (m) | Run-out (m) |
|---|---|---|---|---|---|---|---|---|---|
| 9 | 0.0011 | 0 | 0.8 | 0.9 | 1.3 | 2187.1 | 3.4 | 1.5 | 517.7 |
| 62 | 0.0012 | 9 | 0.9 | 1.0 | 1.4 | 1394.7 | 3.3 | 1.5 | 533.8 |
| 64 | 0.0012 | 14 | 0.8 | 0.8 | 1.1 | 2106.1 | 3.5 | 1.6 | 519.0 |
| 65 | 0.0012 | 14 | 0.8 | 0.9 | 1.3 | 2006.9 | 3.5 | 1.5 | 517.7 |
| 66 | 0.0012 | 14 | 0.8 | 1.0 | 0.9 | 1862.7 | 3.5 | 1.5 | 517.7 |
| 82 | 0.0013 | 9 | 0.9 | 0.9 | 1.1 | 2184.5 | 3.5 | 1.6 | 517.7 |
| 88 | 0.0013 | 14 | 0.9 | 0.8 | 1.2 | 1391.7 | 3.2 | 1.5 | 533.8 |
| 90 | 0.0013 | 14 | 0.9 | 1.0 | 1.7 | 1450.9 | 3.3 | 1.5 | 533.8 |
| 93 | 0.0013 | 21 | 0.8 | 0.9 | 1.0 | 2333.0 | 3.5 | 1.7 | 521.3 |
| 94 | 0.0013 | 21 | 0.8 | 1.0 | 1.2 | 2245.5 | 3.5 | 1.7 | 521.3 |
| 117 | 0.0014 | 21 | 0.9 | 0.9 | 1.1 | 2041.5 | 3.5 | 1.5 | 517.5 |
| 118 | 0.0014 | 21 | 0.9 | 1.0 | 1.2 | 1336.6 | 3.4 | 1.5 | 533.8 |

Models of DF#1 are highlighted in blue and of DF#2 in orange



**Table 5.** Statistics summary for scenarios A, B and C

| Scenarios | Excess rain (mm h) | Max. Vel. (m s) | Max. Thickn. (m) | Total volume (m³) | Max. Run-out (m) |
|---|---|---|---|---|---|
| A | 30 | 3.5 | 4.2 | 15,231 | 734 |
| B | 35 | 6.8 | 6.6 | 415,078 | 1849 |
| C | 40 | 7.5 | 16.6 | 1,038,710 | 2467 |

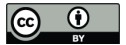

**Table 6.** Classification of the deposits thickness and occupied area (Scenario A)

| Thickness classes of the deposits (m) | | | |
|---|---|---|---|
| [0 – 0.5[ | Area (m²) | | 73,275 |
| | % | | 92.3 |
| [0.5 – 1.0[ | Area (m²) | | 2900 |
| | % | | 3.6 |
| [1.0 – 2.0[ | Area (m²) | | 2425 |
| | % | | 3.1 |
| [2.0 – 3.0[ | Area (m²) | | 625 |
| | % | | 0.8 |
| [3.0 – 4.2] | Area (m²) | | 125 |
| | % | | 0.2 |





**Table 7.** Classification of the deposits thickness and occupied area (Scenario B)

| Thickness classes of the deposits (m) | | | |
|---|---|---|---|
| | [0 – 0.5[ | Area (m²) | 1,180,300 |
| | | % | 85.6 |
| | [0.5 – 1.0[ | Area (m²) | 116,375 |
| | | % | 8.4 |
| | [1.0 – 2.0[ | Area (m²) | 59,625 |
| | | % | 4.3 |
| | [2.0 – 3.0[ | Area (m²) | 18,650 |
| | | % | 1.3 |
| | [3.0 – 4.0[ | Area (m²) | 3650 |
| | | % | 0.3 |
| | [4.0 – 5.0[ | Area (m²) | 400 |
| | | % | 0.03 |
| | >= 5 | Area (m²) | 100 |
| | | % | 0.007 |





**Table 8.** Classification of the deposits thickness and occupied area (Scenario C)

| | | | |
|---|---|---|---|
| **Thickness classes of the deposits (m)** | [0 – 0.5[ | Area (m²) | 4,056,375 |
| | | % | 88.6 |
| | [0.5 – 1.0[ | Area (m²) | 276,000 |
| | | % | 6.0 |
| | [1.0 – 2.0[ | Area (m²) | 157,150 |
| | | % | 3.4 |
| | [2.0 – 3.0[ | Area (m²) | 59,925 |
| | | % | 1.3 |
| | [3.0 – 4.0[ | Area (m²) | 16,250 |
| | | % | 0.4 |
| | [4.0 – 5.0[ | Area (m²) | 5800 |
| | | % | 0.1 |
| | >= 5 | Area (m²) | 8150 |
| | | % | 0.2 |



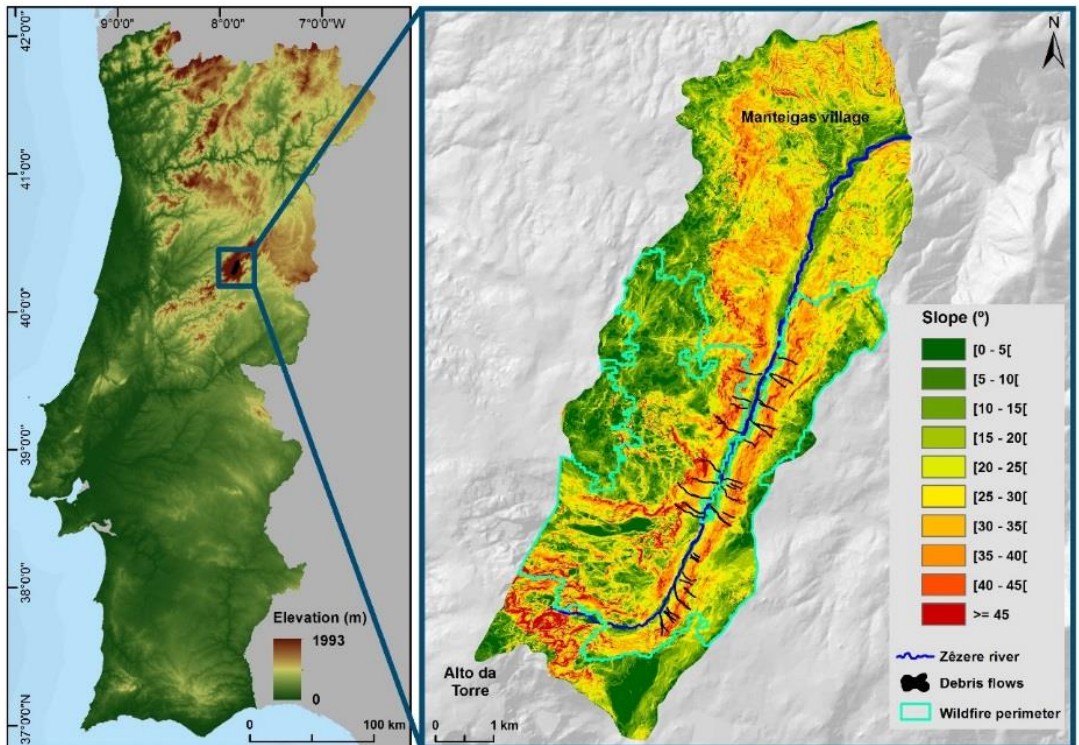

**Figure 1.** Location of the study area and debris flows inventory.




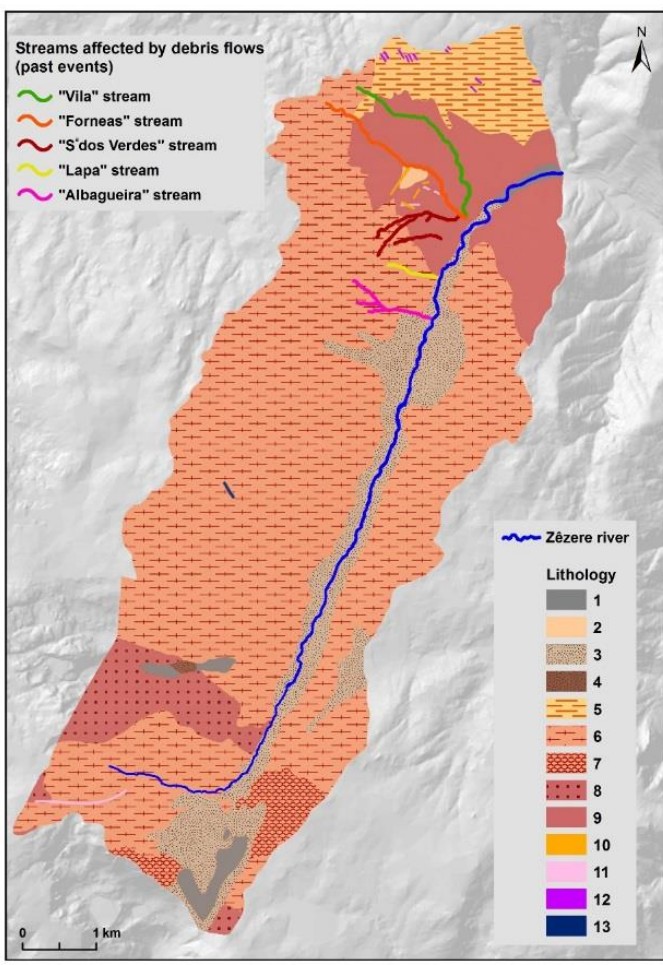

**Figure 2.** Lithology of the study area and streams affected by historical occurrences of debris flows.
Lithology: 1 = alluvium; 2 = slope deposits; 3 = fluvioglacial deposits; 4 = glacial deposits; 5 = contact metamorphic rock (hornfels); 6 = porphyritic medium- to coarse-grained two-mica granite; 7 = porphyritic medium-grained two-mica granite; 8 = non-porphyritic medium- to coarse-grained muscovite granite; 9 = non-porphyritic medium- to fine-grained biotite granite; 10 = quartz dikes; 11 = basic rock dikes; 12 = metamorphosed basic dikes; 13 = aplite-pegmatite dikes.




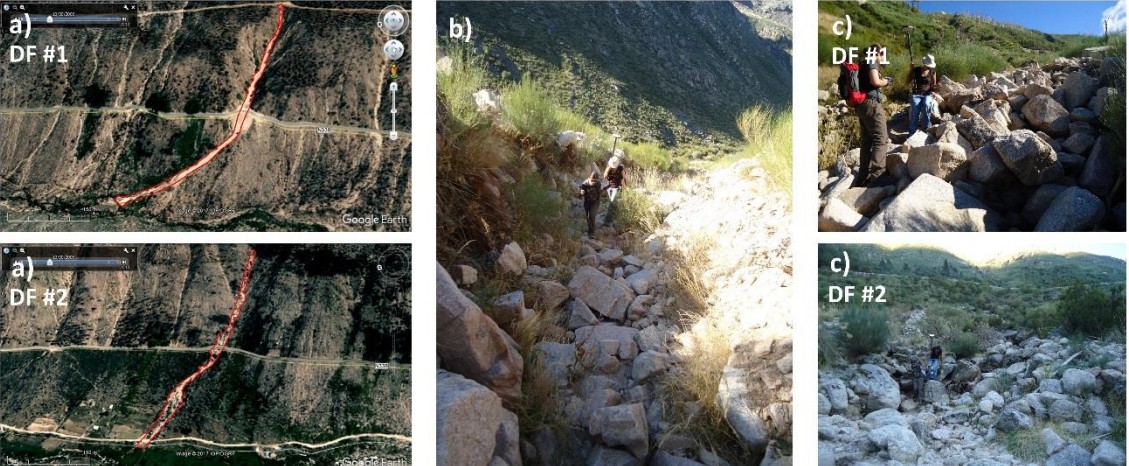

**Figure 3. (a)** Debris flows identification and delimitation; **(b)** Material deposited in the transport zone; **(c)** Deposits in the accumulation area.



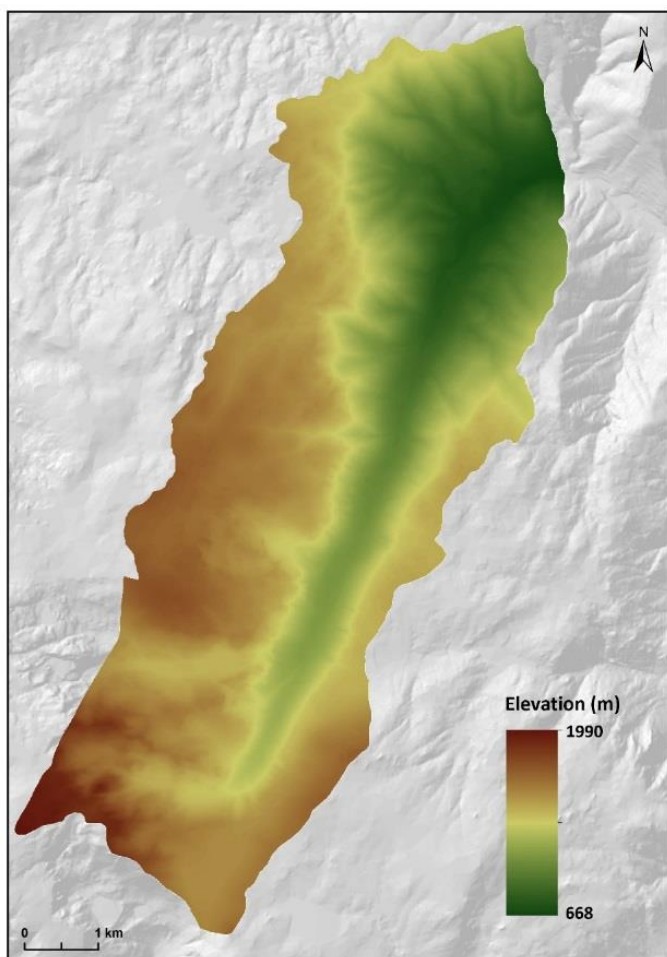

**Figure 4.** Digital elevation model (DEM) of the study area.





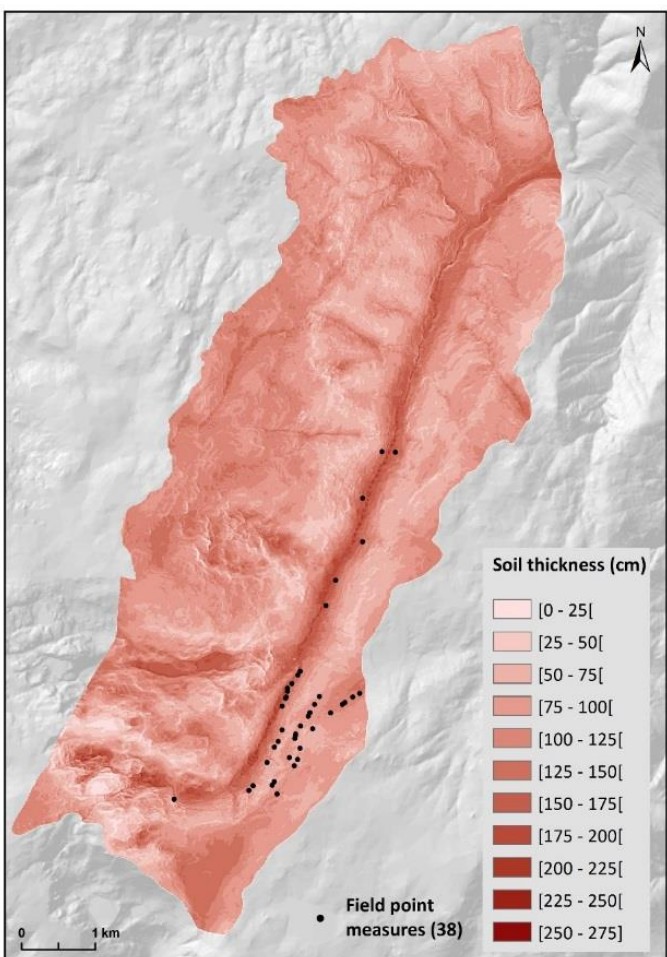

**Figure 5.** Map of the soil thickness of the study area.



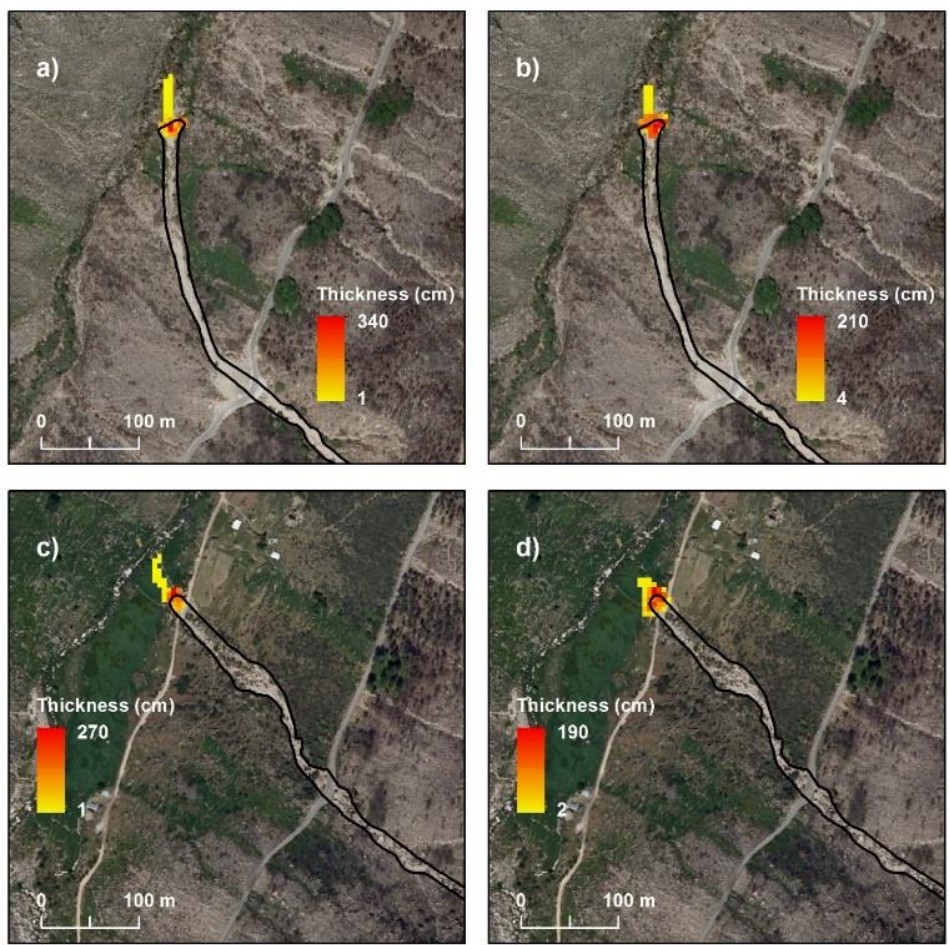

**Figure 6.** DF#1 and DF#2 modeling with the Coulomb frictional **(a, c)** and with the Voellmy **(b, d)** models.




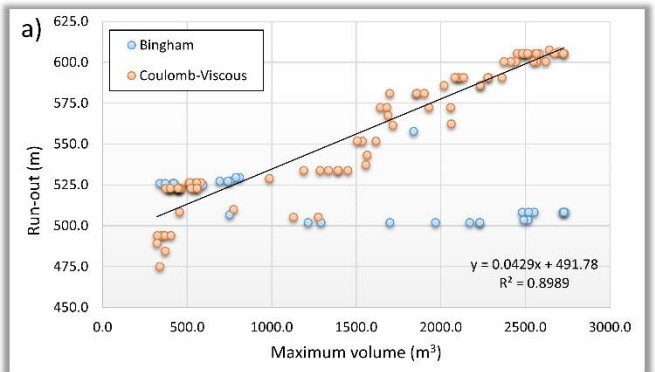
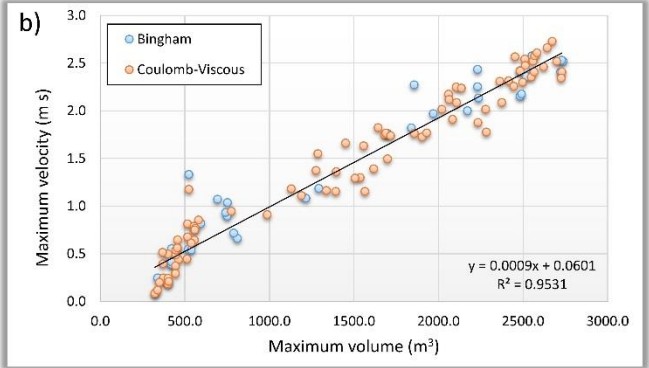

**Figure 7.** Correlation between the maximum volume and **(a)** the run-out and **(b)** the maximum flow velocity, for DF#1.





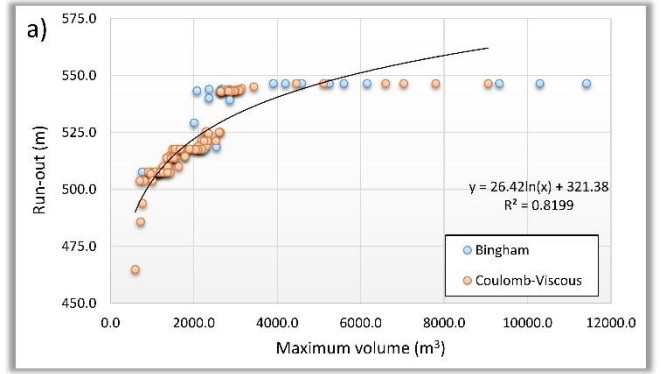
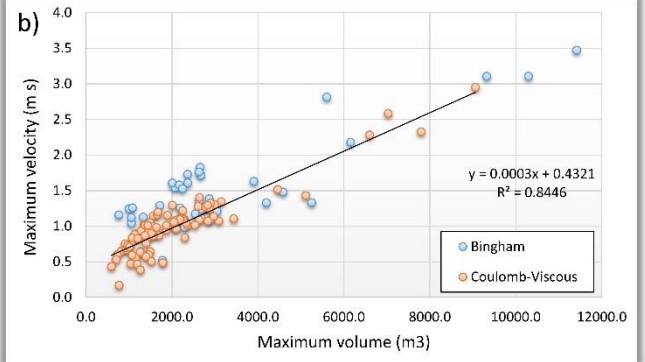

**Figure 8.** Correlation between the maximum volume and **(a)** the run-out and **(b)** the maximum flow velocity, for DF#2.





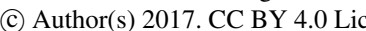

5 **Figure 9.** Temporal evolution of the deposited material thickness considering an accurate model (Model run #62 for DF#1 and Model run #64 for DF#2).





**Figure 10.** Debris flows run-out modeling at the basin scale, considering an excess rain of 30 mm h$^{-1}$ (Scenario A).



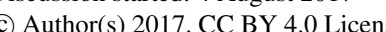

**Figure 11.** Debris flows run-out modeling at the basin scale, considering an excess rain of 35 mm h$^{-1}$ (Scenario B).



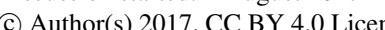

**Figure 12.** Debris flows run-out modeling at the basin scale, considering an excess rain of 40 mm h$^{-1}$ (Scenario C).