# Peer review of "Debris flow run-out simulation and analysis using a dynamic model"

_Natural Hazards and Earth System Sciences, 2017_

## Referee Comment (RC1) · Anonymous Referee #1 · 17 Sep 2017

general comments The manuscript titled "Debris flow run-out simulation and analysis using a dynamic model", by Melo et al., concerns an interesting topic related to df scenarios and potential damage in a study basin of Central Portugal. Taking into account the topic of the special issue, some comments on possible applications in a warning system should be included. The employed model is not new. The manuscript describes an example of application of the model to map zones susceptible to debris flows originated from hydraulic erosion. Unfortunately, only scarce information seems to be available on the real cases selected for calibration/validation. Therefore, limited chances of true calibration/validation exist, and obtained scenarios seem to have a limited practical significance. Nevertheless, the manuscript could be considered for publication after moderate/major revisions. More details on application of the model to simulate real

cases during calibration and validation phases should be given to better understand input data, assumptions and model initialization, and adopted criteria of evaluation of the results - from sites of initiation to travel paths to deposition zones. Further discussion would be important on the following issues: significance of rain data with respect to sites of debris flow development, and of adopted excess rain values, erosion coefficient and rheological parameters; available information on initiation-track-deposition zones of real cases, on depth of erosion due to entrainment, on thickness of deposits, on velocities attained. References should be extended by mentioning other types of modelling approaches.

detailed comments page 2 line 2: perhaps, it would be better to generalize "without past or historical occurrences" into "without past known occurrences". lines 8-23: actually, there are also other types of models, suitable to medium and small scale applications, based on a discretized viewpoint (e.g. cellular automata) of the considered slopes, and on either empirical or semi-empirical approaches. Initial soil slip and successive entrainment of further material along the path of the flow are considered therein by a combination of elementary processes, acting within the cells of the computational domain. Despite adopting simplified approaches (e.g. the equivalent fluid), the rheological issues are also taken into account in some of these models through energy-dissipation options. Authors may here add some reference to some of such papers available in literature. Moreover, some comments should be included on techniques commonly adopted in literature to overcome difficulties in model calibration and validation. The need of quantitative, automated and exhaustive evaluations and sensitivity analyses (e.g. based on genetic algorithms) should also be stressed.

page 3 section 2 (Study area) Based on types of phenomena mainly considered in the paper (cf. shallow landsliding and erosion), some information on weathering conditions of the outcropping terranes would be useful. lines 20-21: please improve the sentence ("generated 20 dead people" sounds quite bad..).

page 4 line 22: some considerations should be included on all assumptions made on

all model parameters (e.g. here, critical height for erosion) and other considered factors (rainfall, soil thickness, etc.).

page 5 line 5: please check and improve the sentence "The routing of solids and water are separately, obeying". lines 12-13: please check and improve the sentence "These two debris flows were selected based on its size and volume of the mobilized material, as well as the conservation of deposits at the time of field surveying.". line 15: it is not clear whether the overlap between simulated and real case was considered or not for calibration and validation purposes (cf. criteria a-c, lines 15-19). Such type of evaluations should be performed, whenever feasible, in a quantitative way by employing a suitable fitness function (as commonly seen in literature). lines 22-23: the sentence "It is also known that rainfall hourly data are more important than rainfall cumulative daily data." is correct if restricted to shallow landslides and hydraulic processes on the slopes. lines 30-31 (and following): the lack of information on amounts of rainfall and on characteristics of the soils in the study area seems a limiting issue for proper model applications. Authors should extend the discussion of such limitations and of solutions adopted.

page 8 lines 16-17: "We found out that excess rain values lower than 28 mm h-1 do not generate debris flows. Thus, the calibration was performed using an amplitude of 1 mm h-1." it is not clear how such threshold was determined. Please, add some more details and explain better the connection between the two sentences. lines 17-22: again, please give more details on how mentioned thresholds were determined.

page 9 line 6: "..related with the run-out.." perhaps "related to"?

page 10: lines 8-9: "..according to the models performed, it is possible to determine that the stages of initiation, transport and deposition of DF #1 had a total duration of 53 minutes, whereas DF#2 occurred over a period lasting 77 minutes." I'd rather say that these durations result from model applications. A description of what actually occurred in the field is not so immediate.. line 16: "In a first analysis, the values obtained for

scenarios A and B appear to be realistic.." how was performed such evaluation? did you use an objective criterion to decide when they are not realistic? (even considering limitations on field data) lines 18- end of section 4: before commenting presence of buildings in areas affected by the simulated flows, you should discuss (in quantitative way) the ability of the model to simulate known real cases (by comparing simulated vs. real affected areas, as said above). Moreover, once calibrated against real cases, the model should be validated (again, in a quantitative way) against further real cases (not employed for calibration). It is not clear whether/how calibration and validation was performed in this study. section 5 (concluding remarks) To be updated after revisions to the other sections.

---

## Referee Comment (RC2) · Anonymous Referee #2 · 13 Oct 2017

Given the relatively limited number of studies that apply dynamic models over larger areas, at a medium scale , this paper gives a very interesting contribution. It analyses several debris flows that were triggered after a wildfire, and uses the parameters for modelling debris flows at a basin scale under different scenarios. Given the basin scale application the authors use a simple model for evaluating the required excess rain and for analysing the entrainment, and use fixed a threshold for sediment concentration, and a simplified soil thickness model. Although these could be debated, their selection is explainable given the scale of analysis and the lack of data. The paper is well written and the results are clearly presented. I recommend that the paper is published as it is. It is interesting to note that the study area bears the same name as one of the co-authors. . .

---

## Author Comment (AC1) · 18 Dec 2017

We are very thankful for the appreciation made by the Anonymous Reviewer #2.
* * *

---

## Author Comment (AC2) · 18 Dec 2017

The manuscript titled "Debris flow run-out simulation and analysis using a dynamic model", by Melo et al., concerns an interesting topic related to df scenarios and potential damage in a study basin of Central Portugal.

**Authors reply:** we are very thankful for the appreciation made by the Anonymous Reviewer #1. All the comments and suggestions were separated in topics and answered one by one.

1) Taking into account the topic of the special issue, some comments on possible applications in a warning system should be included.

**Authors reply:** We acknowledge the request raised by the reviewer, which was also previously mentioned by the editor and suggested to be discussed during this stage.
In addition, the editor asked us to provide information about computational time and minimum system requirements. To answer these issues, the following comments will be included in the new version of the manuscript (section 4, Results and Discussion):

The model was implemented in the PCRaster environmental modeling language, which runs on Linux and Windows. Concerning this work, we used a computer with a i3 2.13 GHz processor and 4 GB of RAM (64-bit operating system). Under these system specifications, the debris flows simulation at the basin scale (over 44 km$^2$) took ca. 72 hours to be completed. Although a faster performance can be obtained using more powerful machines, this dynamic model is not appropriate to be used, as it is, for real-time early warning systems. Even a better computational time would unlikely surpass the gap between the rainfall event and the occurrence of debris flows initiated by surface runoff. In this study we used the excess rain values to calibrate the model given the lack of information about the rainfall amount and the soils in the catchments. This option is a good first approach to define hazardous areas, as well as the maximum volume and velocity reached by debris flows. Whenever a nearby rain gauge is available, a deeper investigation about the characteristics of the soils should be considered, at least on the most hazardous sites, in order to use the hydraulic conductivity values to develop a wide range of scenarios at the basin scale using different precipitation inputs. In case of imminent risk, the most suitable scenario can be chosen according to the amount of rainfall at that time. This information can be immediately directed to civil protection and first-response emergency services.

2) The employed model is not new. The manuscript describes an example of application of the model to map zones susceptible to debris flows originated from hydraulic erosion. Unfortunately, only scarce information seems to be available on the real cases selected for calibration/validation. Therefore, limited chances of true calibration/validation exist, and obtained scenarios seem to have a limited practical significance.

**Authors reply:** We agree the model is not new although it has been used in a scientific publication one single time and for a local (slope) scale. We further agree the available information is scarce, but this is a major challenge of our work, i.e., how to apply a dynamic propagation model using limited input data.
A new validation step will be included in the new version of the manuscript as described in the following reply. We consider this will strength the practical significance of the used scenarios.

3) **More details on application of the model to simulate real cases during calibration and validation phases should be given to better understand input data, assumptions and model initialization, and adopted criteria of evaluation of the results - from sites of initiation to travel paths to deposition zones.**

**Authors reply:** We acknowledge the modeling strategy description and assumptions were not enough clear in the previous version of the manuscript. This will be improved in the new version of the manuscript, namely by including an intermediate step devoted to validation. In the first step we use the two largest debris flows to calibrate the model; in the second step we validate the model considering 32 debris flows not used in calibration; and in step 3 we apply the model to the basin using three realistic scenarios.
Due to data limitations, the evaluation of results is only possible using the spatial agreement. We think this will be enough clear in the new version of the manuscript.

4) **Further discussion would be important on the following issues: significance of rain data with respect to sites of debris flow development, and of adopted excess rain values, erosion coefficient and rheological parameters; available information on initiation-track-deposition zones of real cases, on depth of erosion due to entrainment, on thickness of deposits, on velocities attained.**

**Authors reply:** We acknowledge the suggestion of the reviewer, and a new piece of text will be included in the concluding remarks section:

If more precise information exists for the study area, namely the characteristics of the soils in the catchments and the hourly total amount of rainfall obtained near the sites of debris flow development, the model would use these parameters instead of being calibrated based on excess rain. This would allow estimating critical rainfall thresholds for the study area (Van Asch et al., 2014). In addition, further information on initiation-track-deposition zones of real cases, on the depth of erosion due to entrainment, on the thickness of deposits, and on velocities attained by real debris flows would reduce the uncertainty of all assumptions that were made when assigning the model parameters and would significantly increase the reliability of calibration and validation of the model.

5) **References should be extended by mentioning other types of modelling approaches.**

**Authors reply:** This will be done in Introduction section.

**detailed comments:**

6) **page 2 line 2: perhaps, it would be better to generalize "without past or historical occurrences" into "without past known occurrences".**

**Authors reply:** We agree with the reviewer, thus the sentence will be replaced as suggested.

7) **lines 8-23: actually, there are also other types of models, suitable to medium and small scale applications, based on a discretized viewpoint (e.g. cellular automata) of the considered slopes, and on either empirical or semi-empirical approaches. Initial soil slip and successive entrainment of further material along the path of the flow are considered**

**therein by a combination of elementary processes, acting within the cells of the computational domain. Despite adopting simplified approaches (e.g. the equivalent fluid), the rheological issues are also taken into account in some of these models through energy-dissipation options. Authors may here add some reference to some of such papers available in literature. Moreover, some comments should be included on techniques commonly adopted in literature to overcome difficulties in model calibration and validation. The need of quantitative, automated and exhaustive evaluations and sensitivity analyses (e.g. based on genetic algorithms) should also be stressed.**

**Authors reply:** We acknowledge the considerations made by the reviewer and we will improve the section 1 (introduction) of the manuscript by including some comments. To overview where these comments will be included, we transcribe part of the manuscript where the referred comments are highlighted in blue, as well as the references.

The dynamic approach is numerically solved through physically-based models based on fluid mechanics. Dynamic continuum models rely on the application of conservation laws of mass, momentum and energy, and the behavior of the flow material is defined by its rheological properties (Dai et al., 2002; Quan Luna et al., 2012). Some of the rheological models frequently used to describe the behavior of debris flows are (e.g., Naef et al., 2006; Beguería et al., 2009; Hungr and McDougall, 2009): 1) the Coulomb frictional resistance; 2) the Bingham model; 3) the Coulomb-Viscous; and (4) the Voellmy frictional-turbulent resistance.

One of the major challenges of the dynamic approach is to select the most appropriate rheology to simulate the flow behavior as well as to correctly estimate or calibrate the model's key parameters (e.g., O'Brien et al., 1993; Beguería et al., 2009; Hsu et al., 2010; Scheidl et al., 2013). Most frequently, these parameters are estimated by back-analysis of past events (e.g. Naef et al., 2006; Rickenmann et al., 2006; Hürlimann et al., 2008). The calibration by back-analysis is usually performed by *trial and error*. However, sophisticated automated techniques, like the application of genetic algorithms (e.g. Iovine et al., 2005; D'Ambrosio et al., 2006; Spataro et al., 2008; Terranova et al., 2015) has made possible to obtain more exhaustive evaluations of the calibration parameters.

Considering that dynamic models rely on physical-based assumptions, in most cases the validation is a complex task. For instance, if the landslide total volume deposited is known, this information can be used for validation purposes (e.g. Van Asch et al., 2014). More often, only scarce information is available and the validation is carried out through the comparison between the spatial pattern of the landslide simulations and the real cases. Frequently, fitness functions (e.g. Iovine et al., 2005; D'Ambrosio et al., 2006; D'Ambrosio and Spataro, 2007; Spataro et al., 2008; Avolio et al., 2013; Lupiano et al., 2015) has been implemented to perform the spatial comparison of real and simulated events.

Consequently, although widely applied at the slope scale, the dynamic approach has been scarcely applied at the medium scale. Regarding the latter, it stands out the work developed by Revellino et al. (2004) and Hürlimann et al. (2006) that applied one-dimensional numerical models. More recently, Quan Luna et al. (2016) have implemented the "AschFlow", a two-dimensional one-phase continuum model that simulates the debris flow erosion and deposition processes. In this model, the debris flows are initiated by soil slips and the flow behavior is conditioned by the rheology.

It is worth mentioning other type of dynamic models, for instance based on a discretized viewpoint (e.g. cellular automata), that are suitable to medium and small scale analysis (e.g. D'Ambrosio et al., 2003; Iovine et al., 2003; Avolio et al., 2011, 2013; Lupiano et al., 2015). Such models integrate the initial soil slip and further material entrainment along the path by a combination of elementary processes, acting within the cells of the computational domain. Despite adopting simplified approaches (e.g. the equivalent fluid), the rheological issues are also taken into account through energy-dissipation options.

**REFERENCES:**

Avolio, M.V.; Bozzano, F.; D'Ambrosio, D.; Di Gregorio, S.; Lupiano, V.; Mazzanti, P.; Rongo, R.; Spataro, W. (2011). Debris flows simulation by celular automata: a short review of the SCIDDICA models. Italian Journal of Engineering Geology and Environment, p. 387–397.

Avolio, M.V.; Di Gregorio, S.; Lupiano, V.; Mazzanti, P. (2013). SCIDDICA-SS$_3$: a new version of cellular automata model for simulating fast moving landslides. J Supercomput, 65:682–696.

D'Ambrosio, D.; Di Gregorio, S.; Iovine, G. (2003). Simulating debris flows through a hexagonal cellular automata model: SCIDDICA S$_{3-hex}$. Natural Hazards and Earth System Sciences, 3: 545–559.

D'Ambrosio, D.; Spataro, W.; Iovine, G. (2006). Parallel genetic algorithms for optimising cellular automata models of natural complex phenomena: An application to debris flows. Computers & Geosciences, 32: 861–875.

D'Ambrosio, D.; Spataro, W. (2007). Parallel evolutionary modelling of geological processes. Parallel Computing, 33(3): 186-212.

Iovine, G.; Di Gregorio, S.; Lupiano, V. (2003). Debris-flow susceptibility assessment through cellular automata modeling: an example from 15–16 December 1999 disaster at Cervinara and San Martino Valle Caudina (Campania, southern Italy). Natural Hazards and Earth System Sciences, 3: 457–468.

Iovine, G.; D'Ambrosio, D.; Di Gregorio, S. (2005). Applying genetic algorithms for calibrating a hexagonal cellular automata model for the simulation of debris flows characterized by strong inertial effects. Geomorphology, 66, 287–303.

Lupiano, V.; Machado, G.; Crisci, G.M.; Di Gregorio, S. (2015). A modelling approach with macroscopic cellular Automata for hazard zonation of debris flows and lahars by computer simulations. Int. J. Geol. 9, 35–46.

Spataro, W.; D'Ambrosio, D.; Avolio, M.V.; Rongo, R.; Di Gregorio, S. (2008). Complex Systems Modeling with Cellular Automata and Genetic Algorithms: An Application to Lava Flows. Proceedings of the 2008 International Conference on Scientific Computing, Las Vegas, Nevada, USA.

Terranova, O.G.; Gariano, S.L.; Iaquinta, P.; Iovine, G. (2015). [GA]*SAKe*: forecasting landslide activations by a genetic-algorithms-based hydrological model. Geosci. Model Dev., 8: 1955–1978.

8) **page 3 section 2 (Study area) Based on types of phenomena mainly considered in the paper (cf. shallow landsliding and erosion), some information on weathering conditions of the outcropping terranes would be useful.**

**Authors reply:** We agree with the reviewer and we will include the following text to the manuscript:

In the study area, the effects of weathering on rock strength is mostly evident in coarse and medium granites (Migoń and Vieira, 2014).

Migoń and Vieira (2014) ranked the granite types according to their resistance against weathering and erosion as it follows: non-porphyritic medium- to coarse-grained muscovite granite (lithology 8, Fig. 2) > porphyritic medium- to coarse-grained two-mica granite (lithology 6, Fig. 2) > non-porphyritic medium- to fine-grained biotite granite (lithology 9, Fig. 2).

According to Migoń and Vieira (2014), lithology 6 is less resistant than lithology 8 especially due to the texture, since the mineralogical composition of both lithologies is similar. The authors also observed, from road cuts in steep slopes, that lithology 9 is deeply weathered. Plus, they refer that this condition is increased by the high biotite content favoring chemical weathering, the presence of magmatic foliation and hydrothermal changes.

**9) lines 20-21: please improve the sentence ("generated 20 dead people" sounds quite bad..).**

**Authors reply:** We agree with the reviewer, thus the sentence will be changed as it follows:

The first records on debris flows, occurred in the study area, date back to the 19$^{th}$ century. In a particular event, registered in 1804 (occurred along the stream marked in green, in Fig. 2), 20 lives were lost (Melo and Zêzere, 2017).

**10) page 4 line 22: some considerations should be included on all assumptions made on all model parameters (e.g. here, critical height for erosion) and other considered factors (rainfall, soil thickness, etc.).**

**Authors reply**: We agree with the reviewer suggestion and additional considerations on the assumptions on model parameters will be provided in the new version of the manuscript (see reply #4).

In the particular case of critical height for erosion the following addition information will be given in the new version of the manuscript:

$h*$ is the critical height for erosion to occur which is arbitrarily assigned the value of 0.1 m (Van Asch et al., 2014), thus ignoring the soil erosion associated to lower flow height.

**11) page 5 line 5: please check and improve the sentence "The routing of solids and water are separately, obeying".**

**Authors reply:** The sentence will be changed as it follows:

The solids and water are routed separately, obeying the law of mass conservation, and for each time step a new concentration is calculated (Van Asch et al., 2014).

**12) lines 12-13: please check and improve the sentence "These two debris flows were selected based on its size and volume of the mobilized material, as well as the conservation of deposits at the time of field surveying."**

**Authors reply:** The sentence will be changed as it follows:

These two debris flows were selected because of their largest dimension and good preservation in the landscape.

**13) line 15: it is not clear whether the overlap between simulated and real case was considered or not for calibration and validation purposes (cf. criteria a-c, lines 15-19). Such type of evaluations should be performed, whenever feasible, in a quantitative way by employing a suitable fitness function (as commonly seen in literature).**

**Authors reply:** we agree with all the comments made by the reviewer, since the first version of the manuscript does not demonstrate a proper validation. This will be included in the new version of the manuscript and the new text (in section 3.3, Model Setup) is highlighted in blue. Furthermore, table 4 will be updated with the information regarding the fitness function values obtained for the most accurate simulations, as well as with the percentage of area with a perfect match between the simulated and the real cases. Some comments will also be added to section 4 (Results and discussion).

3.3 Model setup
The calibration of the model was performed for two debris flows triggered during the event of 2005. These two debris flows were selected because of their largest dimension and good preservation in the landscape. Due to the lack of quantitative information for calibration (i.e., thickness and volume of the deposits), we considered that the outputs from the model could be valid if they positively answered all the following criteria (Fig. 3): a) The modeling results must reveal an agreement between the maximum run-out distance simulated and the maximum run-out distance observed; b) The simulation must mimic the deposition of material, with a few centimeters in thickness, observed along the debris flow transport zone; c) The maximum absolute thickness of the deposits in the accumulation area must not exceed 3.5 m and the mean value must be between 1.5 m and 2.0 m, as we registered during field work. The models were calibrated by *trial and error* and considered valid if the three predefined criteria are verified. The valid simulations were evaluated by using a fitness function (e.g. D'Ambrosio et al., 2006; D'Ambrosio and Spataro, 2007; Spataro et al., 2008; Avolio et al., 2013) and by calculating the percentage of overlapping area between the simulation results and the real cases. In the fitness function **(equation 5)**, $R$ and $S$ are, respectively, the cells affected by the real and the simulated events, whereas $m(R \cap S)$ and $m(R \cup S)$ are, respectively, the measure of their intersection and union.

$$f(R,S) = \frac{m(R \cap S)}{m(R \cup S)}$$

The fitness function delivers values between 0 and 1. If $f(R,S) = 0$, it means the real and simulated events are completely disjoint, being $m(R \cap S) = 0$. On the other hand, if $f(R,S) = 1$, the real and simulated events are perfectly overlapped, being $m(R \cap S) = m(R \cup S)$. Values greater than 0.7 can be considered acceptable for two dimensions (Lupiano et al., 2015).

**Table 4.** Parameters used and estimations obtained for the most accurate simulations

| Simulation number | β (m) | φ (°) | Cohes. (kPa) | Visc. (kPa s) | Max. Vel. (m s) | Vol. (m³) | Max. thickn. (m) | Mean thickn. (m) | Run-out (m) | % of overlapped area (simulated vs real) | Fitness function values |
|---|---|---|---|---|---|---|---|---|---|---|---|
| 9 | 0.0011 | 0 | 0.8 | 0.9 | 1.3 | 2187.1 | 3.4 | 1.5 | 517.7 | 77.7 | 0.88 |
| 62 | 0.0012 | 9 | 0.9 | 1.0 | 1.4 | 1394.7 | 3.3 | 1.5 | 533.8 | 77.4 | 0.88 |
| 64 | 0.0012 | 14 | 0.8 | 0.8 | 1.1 | 2106.1 | 3.5 | 1.6 | 519.0 | 77.7 | 0.88 |
| 65 | 0.0012 | 14 | 0.8 | 0.9 | 1.3 | 2006.9 | 3.5 | 1.5 | 517.7 | 77.7 | 0.88 |
| 66 | 0.0012 | 14 | 0.8 | 1.0 | 0.9 | 1862.7 | 3.5 | 1.5 | 517.7 | 77.7 | 0.88 |
| 82 | 0.0013 | 9 | 0.9 | 0.9 | 1.1 | 2184.5 | 3.5 | 1.6 | 517.7 | 78.7 | 0.89 |
| 88 | 0.0013 | 14 | 0.9 | 0.8 | 1.2 | 1391.7 | 3.2 | 1.5 | 533.8 | 78.2 | 0.88 |
| 90 | 0.0013 | 14 | 0.9 | 1.0 | 1.7 | 1450.9 | 3.3 | 1.5 | 533.8 | 77.4 | 0.88 |
| 93 | 0.0013 | 21 | 0.8 | 0.9 | 1.0 | 2333.0 | 3.5 | 1.7 | 521.3 | 79.1 | 0.89 |
| 94 | 0.0013 | 21 | 0.8 | 1.0 | 1.2 | 2245.5 | 3.5 | 1.7 | 521.3 | 79.1 | 0.89 |
| 117 | 0.0014 | 21 | 0.9 | 0.9 | 1.1 | 2041.5 | 3.5 | 1.5 | 517.5 | 79.8 | 0.89 |
| 118 | 0.0014 | 21 | 0.9 | 1.0 | 1.2 | 1336.6 | 3.4 | 1.5 | 533.8 | 77.4 | 0.88 |

Models of DF#1 are highlighted in blue and of DF#2 in orange

**4. Results and discussion**

The evaluation of the 12 most accurate models (table 4) demonstrates that in every single case the overlapping area between the simulation and the real case is above 77%. Furthermore, the fitness function values obtained in all the simulations considered show a good spatial agreement between the modeling results and the real debris flows.

**14) lines 22-23: the sentence "It is also known that rainfall hourly data are more important than rainfall cumulative daily data." is correct if restricted to shallow landslides and hydraulic processes on the slopes.**

**Authors reply:** We totally agree with the reviewer. The sentence will be changed as it follows:

It is also known that rainfall hourly data are more important than rainfall cumulative daily data for the triggering of shallow landslides and for the development of hydraulic processes on the slopes.

**15) lines 30-31 (and following): the lack of information on amounts of rainfall and on characteristics of the soils in the study area seems a limiting issue for proper model applications. Authors should extend the discussion of such limitations and of solutions adopted.**

**Authors reply:** We agree with the reviewer suggestion. The sentence will be changed as it follows:

The lack of information on amounts of rainfall and on detailed characteristics of the soils in the study area is a major drawback for proper dynamic model application. To overcome this limiting issue, we chose to calibrate the model based on the excess rain values, which means the amount of water that will feed the surface runoff once the infiltration is exceeded. Thus, surface runoff reflects not only the total rainfall, but also the interaction between rainfall and the soil system.

16) **page 8 lines 16-17: "We found out that excess rain values lower than 28 mm h-1 do not generate debris flows. Thus, the calibration was performed using an amplitude of 1 mm h-1." it is not clear how such threshold was determined. Please, add some more details and explain better the connection between the two sentences.**

17) **lines 17-22: again, please give more details on how mentioned thresholds were determined.**

**Authors reply:** Concerning comments 16) and 17), we agree with the reviewer and we think that the paragraph mentioned was not clear enough in the first version of the manuscript. Hence, it will be changed as it follows (the new text is highlighted in blue):

The calibration of the rheology and the erosion coefficient are only performed after a minimum value of excess rain with capacity to trigger debris flows is determined for the study area. This excess rain threshold was calibrated by manually assigning an initial low value and, based on the modeling results, increasing it (using and amplitude of 1 mm h$^{-1}$) until an excess rain capable to trigger debris flows was found. The *trial and error* calibration demonstrated that excess rain values lower than 28 mm h$^{-1}$ are not enough to mobilize the loose sediments available on the slopes of the study area. By increasing 1 mm h$^{-1}$ of excess rain at each model simulation we found out that values above 30 mm h$^{-1}$ are required for sediment mobilization in DF#1 and DF#2. Regarding the modeling results for the two debris flows (DF#1 and DF#2, see Fig. 3), the *trial and error* calibration have shown that DF#1 requires an excess rain of 32 mm h$^{-1}$ to allow the calibration of the rheology and the erosion factor, since lower values result in insufficient deposit thickness and insufficient run-out distances, despite the variation of the previously mentioned parameters. For the same reasons, DF#2 requires an excess rain of 33 mm h$^{-1}$ to allow the calibration of the rheology and the erosion factor.

18) **page 9 line 6: "..related with the run-out.." perhaps "related to"?**

**Authors reply:** We agree with the reviewer, thus the sentence will be replaced as suggested.

19) **page 10: lines 8-9: "..according to the models performed, it is possible to determine that the stages of initiation, transport and deposition of DF #1 had a total duration of 53 minutes, whereas DF#2 occurred over a period lasting 77 minutes." I'd rather say that these durations result from model applications. A description of what actually occurred in the field is not so immediate..**

**Authors reply:** We totally agree with the reviewer. The sentence will be changed as it follows:

The model application returns a duration of 53 minutes for DF #1 and 77 minutes for DF#2, considering debris flows initiation, transport and deposition stages.

**20) line 16: "In a first analysis, the values obtained for scenarios A and B appear to be realistic.." how was performed such evaluation? Did you use an objective criterion to decide when they are not realistic? (even considering limitations on field data)**

**Authors reply:** The sentence was based on our empirical knowledge of the study area and on the information provided by the historical records. Since this was only an assumption and we did not use an objective criterion, we will remove the sentence in the new version of the manuscript.

**21) lines 18- end of section 4: before commenting presence of buildings in areas affected by the simulated flows, you should discuss (in quantitative way) the ability of the model to simulate known real cases (by comparing simulated vs. real affected areas, as said above). Moreover, once calibrated against real cases, the model should be validated (again, in a quantitative way) against further real cases (not employed for calibration). It is not clear whether/how calibration and validation was performed in this study.**

**Authors reply:** we acknowledge the reviewer comment. To address the validation of the models computed at the basin scale, we will include the following text and a new table (new table 5) in the manuscript in section 3.3 (model setup) and section 4 (Results and discussion).

3.3. Model Setup
After calibration, the model was validated using the remaining 32 debris flows triggered in 2005 that were not considered for calibration. The validation included the estimation of the overlapping area between the simulation results and the real cases, as well as the fitness function obtained for each simulated vs real debris flow.

4 Results and discussion
Prior to the debris flows run-out modeling at the basin scale using three different scenarios, the ability of the models computed with 32 and 33 mm $h^{-1}$ of excess rain (during a period of 2 hours) was validated for the remaining 32 debris flows triggered in 2005 and not considered for calibration and the results are summarized in table 5. Concerning the modeling computed with an excess rain value of 32 mm $h^{-1}$, the mean percentage and the standard deviation of the overlapped areas (simulated vs real) is ca. 50.2 and 37.5, respectively. When an excess rain of 33 mm $h^{-1}$ is considered, the mean percentage increases (67.7) whereas the standard deviation slightly decreases (30.2). The fitness function validation calculated for the simulation using an excess rain of 32 mm $h^{-1}$ shows that 50% of the debris flows have values above 0.7. If we consider an excess rain of 33 mm $h^{-1}$, 78% of the debris flows have values above the referred threshold.

**Table 5.** Validation of 32 debris flows (not used for calibration) considering the modeling results using 32 and 33 mm h$^{-1}$ of excess rain

| DF# | Excess rain (32 mm h$^{-1}$) | | Excess rain (33 mm h$^{-1}$) | |
|---|---|---|---|---|
| | % of overlapped area (simulated vs real) | Fitness function values | % of overlapped area (simulated vs real) | Fitness function values |
| 1 | 78.4 | 0.9 | 93.2 | 1.0 |
| 2 | 6.7 | 0.3 | 26.7 | 0.5 |
| 3 | 4.1 | 0.2 | 51.4 | 0.7 |
| 4 | 78.9 | 0.9 | 100.0 | 1.0 |
| 5 | 73.8 | 0.9 | 91.3 | 1.0 |
| 6 | 19.6 | 0.4 | 55.7 | 0.8 |
| 7 | 5.5 | 0.2 | 25.5 | 0.5 |
| 8 | 23.3 | 0.5 | 61.7 | 0.8 |
| 9 | 3.6 | 0.2 | 9.1 | 0.3 |
| 10 | 5.3 | 0.2 | 23.7 | 0.5 |
| 11 | 3.3 | 0.2 | 20.0 | 0.5 |
| 12 | 22.6 | 0.5 | 64.5 | 0.8 |
| 13 | 100.0 | 1.0 | 100.0 | 1.0 |
| 14 | 97.1 | 1.0 | 100.0 | 1.0 |
| 15 | 63.2 | 0.8 | 78.9 | 0.9 |
| 16 | 100.0 | 1.0 | 100.0 | 1.0 |
| 17 | 10.3 | 0.3 | 53.4 | 0.7 |
| 18 | 70.0 | 0.8 | 79.2 | 0.9 |
| 19 | 74.6 | 0.9 | 88.4 | 0.9 |
| 20 | 20.2 | 0.5 | 56.7 | 0.8 |
| 21 | 0.0 | 0.0 | 12.5 | 0.4 |
| 22 | 22.1 | 0.5 | 55.9 | 0.8 |
| 23 | 27.4 | 0.5 | 33.3 | 0.6 |
| 24 | 94.5 | 1.0 | 95.7 | 1.0 |
| 25 | 34.2 | 0.6 | 65.8 | 0.8 |
| 26 | 39.8 | 0.6 | 49.6 | 0.7 |
| 27 | 99.4 | 1.0 | 99.4 | 1.0 |
| 28 | 99.1 | 1.0 | 99.1 | 1.0 |
| 29 | 83.8 | 0.9 | 88.6 | 0.9 |
| 30 | 64.6 | 0.8 | 92.3 | 1.0 |
| 31 | 82.4 | 0.9 | 95.9 | 1.0 |
| 32 | 98.9 | 1.0 | 98.9 | 1.0 |
| Mean | 50.2 | 0.6 | 67.7 | 0.8 |
| Max. | 100.0 | 1.0 | 100.0 | 1.0 |
| Min. | 0.0 | 0.0 | 9.1 | 0.3 |
| Std | 37.5 | 0.3 | 30.2 | 0.2 |

**22) section 5 (concluding remarks) To be updated after revisions to the other sections.**

**Authors reply:** The section 5 will be updated taking into account the revisions on the previous sections.